# Carbon Dioxide and Heat Fluxes during Reforestation in the North Caucasus

**Elizaveta Satosina** [1,2,*] , **Nurdin Mamadiev** [1,*] , **Lyubov Makhmudova** [1] **and Julia Kurbatova** [1,2]

1 Department of Ecology and Environmental Management, Institute of Oil and Gas, Grozny State Oil Technical University, Khusein Abubakarovich Isaev Prosp. 100, 364051 Grozny, Russia; mls66@mail.ru (L.M.); kurbatova@sev-in.ru (J.K.)

2 A.N. Severtsov Institute of Ecology and Evolution, Russian Academy of Sciences, Leninskiy Prosp. 33, 119071 Moscow, Russia

* Correspondence: lisan.sat@sev-in.ru (E.S.); justemail552@mail.ru (N.M.)

**Abstract:** Human impact on natural ecosystems has significantly increased in recent decades. As a result, the structure and functioning of ecosystems are seriously altered. This in turn affects regional weather and climate conditions through changes in the radiation, water, and carbon balance of ecosystems. Investigating the process of natural ecosystem restoration after disturbances is an important issue in the context of climate change. During monitoring observations of greenhouse gas (GHG) fluxes in a reforestation area in the Chechen Republic, new experimental data on their seasonal variability were obtained, and their sensitivity to changes in environmental conditions was assessed. Forest restoration and land reclamation are essential components of the low-carbon development and decarbonization strategy of the world economy. Observations of GHG fluxes were conducted in the reclaimed area, which was planted with tree seedlings. One year of eddy covariance flux measurements (May 2022–June 2023) demonstrated that $CO_2$ uptake by the reforestation area in a humid continental climate with hot summers and cold winters is determined by the moisture conditions of the growing season. The cumulative net ecosystem exchange (NEE) for the entire measurement period at the carbon farm was 613.7 gC·m$^{-2}$. The uptake of $CO_2$ by the reforestation area was observed only due to active seedling growth during periods of sufficient soil moisture (May 2023). During this time, total NEE uptake was 48.7 gC·m$^{-2}$. Sensible and latent heat fluxes were also dependent on weather conditions, primarily on incoming solar radiation and moisture conditions. For the successful implementation of climate projects aimed at the creation of artificial forest ecosystems with high $CO_2$ uptake capacity, it is essential to develop appropriate hydro-meliorative measures that ensure a sufficient amount of available soil moisture.

**Keywords:** eddy covariance method; carbon dioxide; methane; reforestation; sensible and latent heat fluxes; radiation balance; carbon landfills

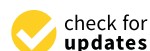



## 1. Introduction

At the moment, the problem of global warming is one of the key concerns in modern society. Most climate experts believe that one of the main reasons for the rise in global temperatures is the increase in the concentration of greenhouse gases (GHGs) in the atmosphere, leading to the intensification of the greenhouse effect [1,2].

Natural ecosystems serve not only as crucial regulators of the Earth's climate system but also represent significant sources and sinks of GHGs. Any changes or disruptions in the structure and functioning of natural ecosystems, whether caused by natural factors such as windfalls, natural fires, or insect pests, or anthropogenic factors like logging and land use changes, undoubtedly lead to alterations in surface albedo, surface roughness, soil moisture regime, evaporation rates, surface and subsurface runoff, and GHG fluxes between the Earth's surface and the atmosphere, thereby directly impacting the radiative,

heat, and hydrological balance of the Earth's surface and, consequently, regional weather and climate conditions [1,3–5].

Until recent years, research on GHG fluxes in reforestation areas has been episodic. Most studies have been conducted either in agricultural areas or clear-cut forests. In previous works [3,4,6], it has been demonstrated that forest logging leads to changes in biogeochemical and biogeophysical processes, significantly influencing energy, water, and carbon dioxide ($CO_2$) fluxes at both local and regional levels, thereby affecting the climate system. Analyzing the temporal variability of $CO_2$ and $H_2O$ fluxes [4] indicates that the most significant flux measurements occur in the first years after forestry activities are carried out. In a study [7], it was shown that land use has a substantial impact on GHG fluxes in tropical forests. Forested areas in the tropics planted for GHG emission compensation can act as sources rather than sinks of GHGs.

Although the issue of reforestation was raised quite a long time ago, it has also been episodically addressed until recently. In a study [8] focused on the carbon balance of temperate ecosystems in China, it was noted that large-scale reforestation programs since the 1980s have turned the southern part of the country into a major carbon sink. Approximately 28%–37% of total fossil carbon emissions since the 1990s have already been absorbed. However, there is a considerable time lag between changes in forest biomass and the implementation of reforestation efforts, which has a destabilizing impact on the overall system dynamics [9].

Understanding the interactions between the Earth's surface and the atmosphere, as well as the impact of plant communities on all atmospheric processes, becomes increasingly relevant in the face of global temperature rise. Therefore, it is essential to actively study the temporal and spatial variability of the radiation, heat, hydrological, and carbon balance of natural ecosystems, as well as their influence on atmospheric greenhouse gas balance. As human influence on ecosystems is growing, comprehensive analysis of the potential consequences of anthropogenic disturbances to these ecosystems on the heat, hydrological, and carbon balance of the Earth's surface is also necessary [3,10]. Investigating the process of natural ecosystem recovery after disturbances and developing optimal restoration techniques for disturbed ecosystems, especially in the context of their role in GHG exchange, is one of the prospective tasks in establishing a global network for monitoring GHG fluxes worldwide.

The aim of this study was to assess the temporal variability of $CO_2$ and latent (LE) and sensible (H) heat fluxes during forest restoration in an anthropogenically disturbed area in the northern part of the Greater Caucasus (Russia, Chechen Republic), using eddy covariance measurements. The specific research objectives were to conduct field observations of $CO_2$, H, and LE using the eddy covariance method; to analyze the representativeness of the selected experimental site; to assess the seasonal and diurnal variability of $CO_2$, H, and LE fluxes; to assess the influence of weather conditions on $CO_2$, H, and LE fluxes. As a working hypothesis, the study suggests that $CO_2$ and heat fluxes in a reforestation area can vary depending on weather conditions, primarily depending on temperature and precipitation.

Reforestation represents one of the most promising strategies for implementing the modern climate agenda by undertaking activities aimed at enhancing the capacity of natural ecosystems to sequester GHG and reduce their emissions into the atmosphere. This research will provide a substantial understanding of the temporal variability of GHG fluxes in reforestation ecosystems. Also, this study will help to improve understanding of the complex interactions between terrestrial ecosystems and the atmosphere.

## 2. Materials and Methods

### 2.1. Site Description

This experimental study was performed at a carbon farm site (43°24′ N and 45°43′ E) (Figure 1). The study area is located in the foothills of the Greater Caucasus, adjacent to the Chechen plain and the Terek–Kuma lowland. The study area also refers to the basin of the Caspian Sea and the middle course of the Terek River.

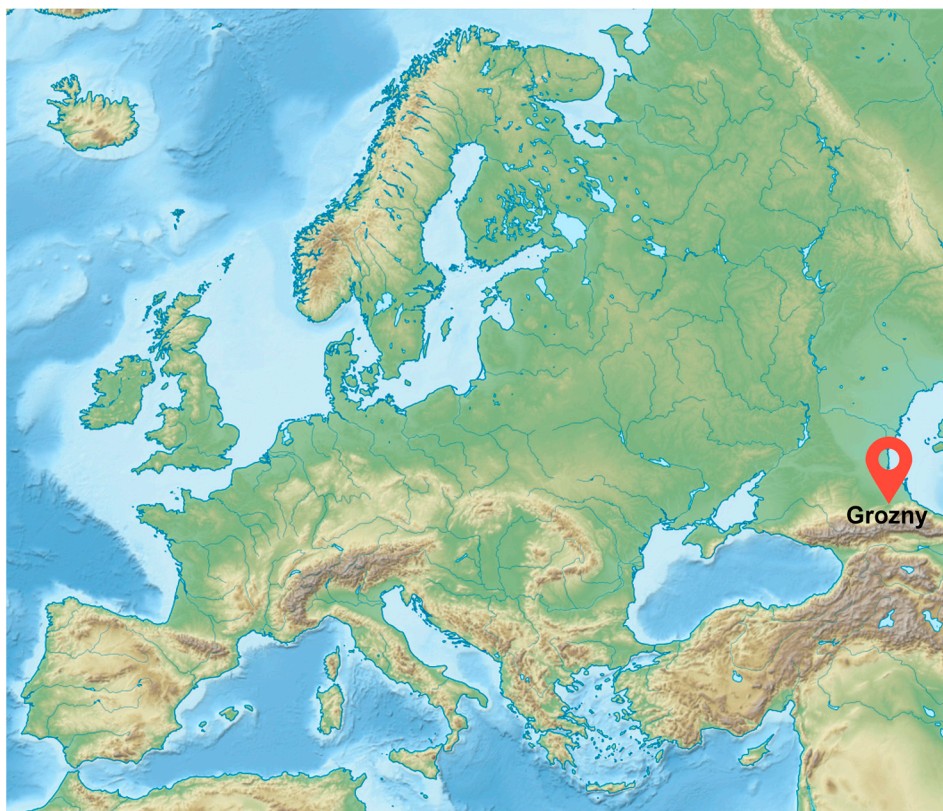

**Figure 1.** Geographical location of the study region in the Chechen Republic.

The region has a humid continental climate (Dfa type in the Köppen–Geiger classification scheme) with hot summers and cold winters [11]. According to the nearest meteorological station in Grozny, the average annual temperature at 2 m height for the period from 1938 to the present is 10.4 °C (−1.5 °C for January and 18.2 °C for July). The average annual precipitation is about 450 mm with a maximum in summer [12]. During the cold season, precipitation falls in the form of snow. Snow cover appears in early December. Usually, it is unstable and during the winter it can melt and reappear several times. In winter there are 45–60 days with snow cover. Its average maximum height does not exceed 10–15 cm. The snow cover disappears by mid-March.

The natural vegetation cover at the carbon farm site is represented mainly by cereal and forb grass on cereal-covered steppes. In the southern part of the belt there are forb–cereal steppes and meadow steppes. In the northern part of the steppe belt, sagebrush and beard-grass groups are common.

Forb–cereal steppes are confined to more humid places with the participation of forbs: *Onobrychis viciifolia* Scop., *Medicago sativa* L., *Trifolium pratense* L., and *Leucanthemum vulgare* Lam. Amongthe cereals, there are *Stipa pennata* L., *Stipa capillata* L., and *Festuca valesiaca Schleich. ex Gaudin*. Arctium settles on the dry slopes, forming the background, creating sagebrush-covered steppes. On the lower terraces in the Sunzha, floodplain broad-leaved forests of the ravine type are widespread. The site is characterized by solonetsous and saline varieties of chestnut soils.

Originally, the area on which the carbon farm is based was a landfill site, where clean-up and reclamation activities were carried out in March 2022. Subsequently, in April 2022, plantings of various deciduous tree species, including *Morus alba* L. (2100 seedlings), *Tilia* L. (1920 seedlings), *Salix alba* L. (2000 seedlings), *Populus alba* L. (520 seedlings), and *Fraxinus excelsior* L. (1936), were conducted. The listed tree species were planted in a mixed pattern at an average distance of about 3 m from each other. The average tree planting density in the study area was 394.2 seedlings per hectare. Due to the absence of irrigation systems and a drought in August–September 2022, nearly all the planted vegetation had dried up

by the end of August 2022. Then, in April 2023, the planting of vegetation at the carbon farm was carried out anew.

### 2.2. Eddy Covariance and Meteorological Measurements

To measure greenhouse gas fluxes at the carbon farm site in the Chechen Republic, measuring equipment standardized for FLUXNET sites was used, including a meteorological station and an eddy covariance station for measuring $CO_2$ flux using the eddy covariance method (Figure 2) [13]. The height of the flux tower is 3.1 m, and it is located in the central part of the carbon farm in an area planted with linden seedlings. The eddy covariance system was installed at the top of the tower and included an enclosed $CO_2/H_2O$ gas analyzer LI-7200RSF (LI-COR Inc., Lincoln, NE, USA) and 3-D ultrasonic Multipath Cage Anemometer with Heater (Metek, Elmshorn, Germany) (Figure 2b).

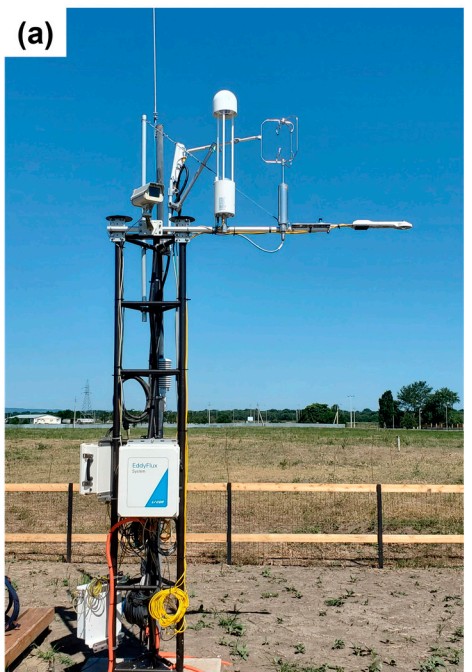 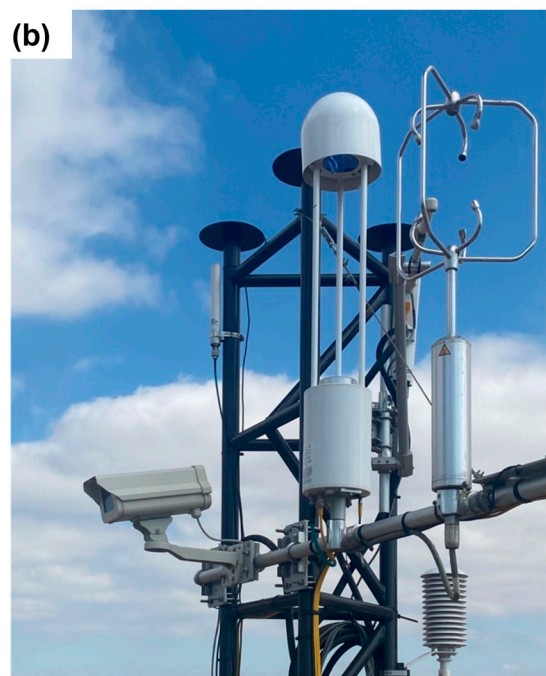

**Figure 2.** The eddy covariance system at the carbon farm site (**a**). The set of instruments installed on the eddy covariance mast (**b**).

Meteorological measurements at the carbon farm were carried out using a MesoPRO (Campbell Sci., Logan, UT, USA) automatic meteorological station for measuring air temperature and humidity, wind speed and direction, amount and duration of precipitation, and incoming solar radiation. A TR-525M rain gauge (Texas Electronics, Dallas, TX, USA) situated at a 1 m height was used to quantify precipitation. For the measurement of short-wave incoming and reflected radiation, as well as long-wave incoming and outgoing radiation, a four-component radiometer CNR4 (Kipp & Zonen B.V., Delft, The Netherlands) was installed at a height of 3 m. A quantum sensor LI-190R (LI-COR Inc., Lincoln, NE, USA) was engaged to monitor the incoming photosynthetically active radiation (PAR). At a depth of 10 cm, three Stevens Hydro Probe II reflectometers (Stevens Water Monitoring Systems Inc., Portland, OR, USA) were used to evaluate soil temperature and soil water volume. Soil heat flux was recorded using three HFP01SC sensors (Hukseflux Thermal Sensors, Delft, The Netherlands) positioned at a 10 cm depth in the mineral soil layer around the flux tower. Eddy covariance details were gathered through an LI-7550 Analyzer interface unit (LI-COR Inc., Lincoln, NE, USA) at a 10 Hz frequency. Extra meteorological data were obtained at a 1 min frequency using a LI-COR Biomet system 103 (LI-COR Inc., Lincoln, NE, USA).

The measurement stations were installed immediately after tree planting in May 2022. For this study, data were used from all measuring systems from the beginning of the measurement period, from 12 May 2022 to 7 June 2023.

### 2.3. Flux Calculation, Footprint and Gap-Filling

The results of eddy covariance measurements were processed in strict accordance with generally accepted recommendations [13,14]. The calculation of H, LE, and $CO_2$ fluxes was carried out for a time interval of 30 min using EddyPro 7 software (LI-COR, Lincoln, NE, USA). The calculations used data on the radiation balance, air and soil temperature, photosynthetically active radiation, air humidity, and other meteorological quantities. The flux values were calculated taking into account all the necessary corrections, which were made in a strict sequence using the EddyPro 7 software package (LI-COR, Lincoln, NE, USA). As the main corrections introduced into the measurements, we considered the rotation of the coordinate system, the acoustic correction for the temperature measured by the anemometer, the correction for the deviation of the direction of the incident air flow from the horizontal, taking into account the signal recording delay (by the covariance maximization method), corrections for the frequency characteristics (restoration in the spectrum of high and low frequencies), correction to eliminate the influence of air density fluctuations, removal of outliers, etc. Quality checking included a 0–2 flag policy [15]. The storage terms of $CO_2$ and heat within the canopy air space were determined as per the guidelines by Migliavacca et al. [16] and Papale et al. [17]. Following the post-processing of the data, all fluxes identified with flag 2, as well as those with flags 0 and 1 that included spikes, which could be linked to occurrences like rain and dew incidents, minimal wind, and weak turbulence, were excluded from the data sets. The data sets of net ecosystem exchange (NEE) were also sorted based on a thorough analysis of the friction velocity (u*) threshold criteria. The threshold values for u*, computed for the entire measurement duration at the carbon farm site, equated to 0.1125 m/s.

For a correct interpretation of the studied fluxes of H, LE, and $CO_2$, it is important that they form within an ecosystem with uniform vegetation and soil cover. This required that the size of the footprint—the area of the earth's surface on the windward side of the measuring tower from which the transfer of energy and matter (substances)—did not exceed the size of the ecosystem under study. The size of footprint depended on the height of the tower and the installed equipment, the stratification of the atmosphere, and the roughness of the underlying surface. As part of the study, the calculation of footprint for the selected area was performed using a two-dimensional parametrization model for predicting the flux footprint (Flux Footprint Prediction, FFP) Kljun et al. [18]. During the selected measurement period, the installation height of the instruments was unchanged, which means that the size of the coverage area was most affected by the surface roughness, which changed during the year depending on the vegetation cover, as well as atmospheric conditions determined by the stratification of the surface air layer.

The peak radius of the coverage area (10% of the total flux) during the measurement period in the daytime for the study area was 90–120 m (Figure 3). The results show that the resulting coverage radius did not exceed the size of the studied carbon farm ecosystem. In the study area, the distance of the maximum contribution (100% of the total flux) in the daytime was about 4–5 m (Figure 3). Thus, in the course of measurements, one of the main conditions for applying the eddy covariance method was fulfilled—all the received fluxes were formed directly within the ecosystem of carbon farm.

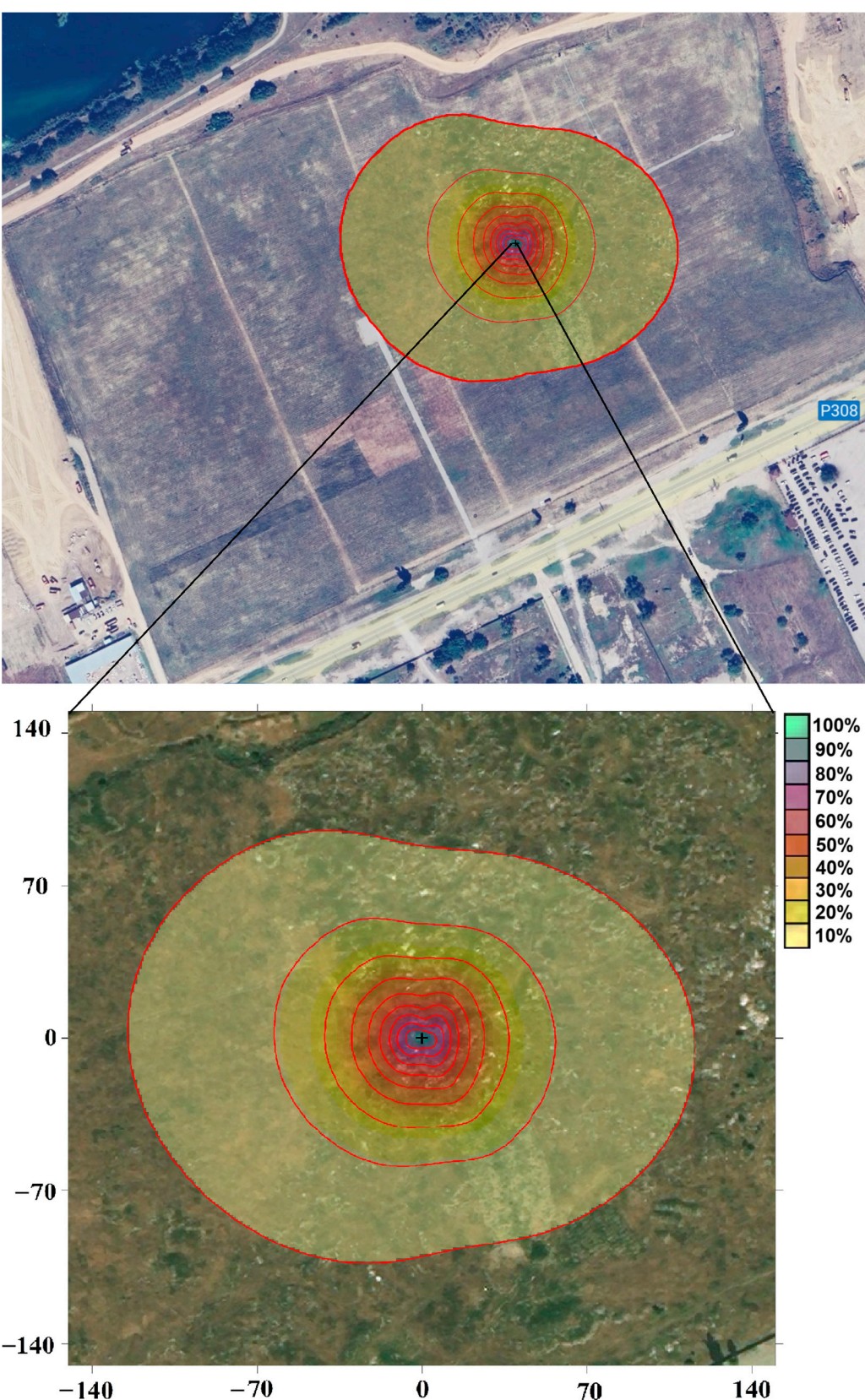

**Figure 3.** Daytime flux footprint based on data collected throughout the entire measurement period. The station's location is marked by a black cross. The red lines delineate areas with different percentage contributions to the measured flux.

The eddy covariance method provides continuous time series of data on the fluxes of H, LE, and $CO_2$ between the earth's surface and the atmosphere. However, after carrying out all the necessary tests and verifying the resulting fluxes, a large number of gaps were present in the observation series (due to unfavorable microclimatic conditions and equipment or power failures), most of which occurred at night or during the autumn–winter season. The percentage of gaps in the data series for H was 51.34%, for LE was 54.77%, and for $CO_2$ was 53.66%. Similar statistics on gaps in the series of these streams has a fairly good percentage of consistency with other eddy covariance stations located in temperate latitudes. The REddyProc packege [19–21] was used in this study for filling the gaps in flux data sets and for deriving NEE partitioning into TER (total ecosystem respiration) and GPP (gross primary production). The $CO_2$ flux between underlying surface and the atmosphere can be represented by the equation:

$$N_E = R_E - GPP$$

where $N_E$ is net ecosystem exchange, $R_E$ is ecosystem respiration, and GPP is gross primary productivity of the photosynthesizing vegetation. In this study, $R_E$ and GPP are presented as positive values, characterizing the absolute magnitudes of ecosystem respiration and $CO_2$ uptake, respectively.

The mean value of energy balance closure for the carbon farm for 30 min intervals was 0.83. The H and LE fluxes were also additionally adjusted for closure using the mean daily β, following Twine et al. [22] and Aubinet et al. [13].

## 3. Results

### 3.1. Meteorological Conditions

During the study period from 12 May 2022 to 7 June 2023, the weather conditions were influenced by the westward transport and the Azores (during the warm period of the year) and the Asian (during the cold period of the year) anticyclones. The measured monthly average air temperature in 2022–2023 was 1.73 °C higher than the long-term mean (averaged for period from 1970 to 2020) (Figure 4). The source of long-term meteorological data from 1970 to 2020 was the All-Russian Research Institute of Hydrometeorological Information—World Data Center (RIHMI-WDC) [12]. Monthly precipitation amounts during the study period varied compared with long-term average values.

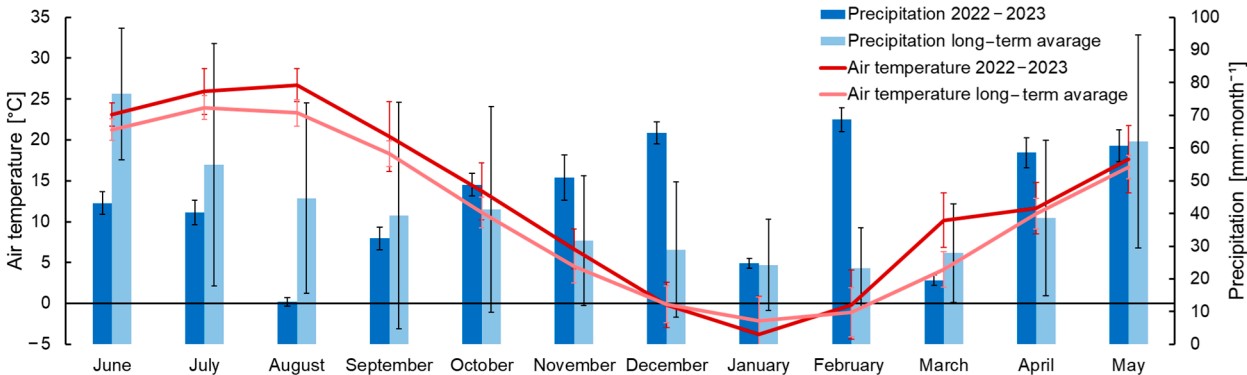

**Figure 4.** Comparison of average monthly temperatures and precipitation for the studied period of 2022–2023 and average monthly long-term values from 1970–2020. Vertical whiskers indicate standard deviation (SD).

The warm period of 2022 was hot and included predominantly dry weather. The daily mean temperature varied from 13.2 to 31.1 °C in the summer of 2022 (Figure 5a). Precipitation in the summer was mainly of a convective nature. The maximum amount of precipitation in 2022 occurred in May and totaled 98 mm for the entire month (Figure 5a). Moreover, during the summer period, precipitation decreased, reaching only 9 mm in Au-

gust. Because of high temperatures (the maximum mean daily temperature was in August and reached 31 °C) (Figure 5a) and a decrease in soil moisture, as well as the complete absence of any irrigation systems at the carbon farm, 100% of all previously planted tree seedlings died. During the autumn, the daily mean air temperatures varied from 28.6 to 1.3 °C. Temperature gradually decreased and reached a minimum of −10.4 °C in mid-January (Figure 5a). Precipitation during the cold period of the year was more frequent, due to increased cyclonic activity, with the maximum amount of precipitation, 38 mm, observed in November. Over the entire measurement period, 613 mm of precipitation fell at the study area. During the winter, precipitation also fell in the form of snow. However, stable snow cover appeared only twice for short times after heavy snowfalls. The maximum depth of snow cover was 8 cm in February.

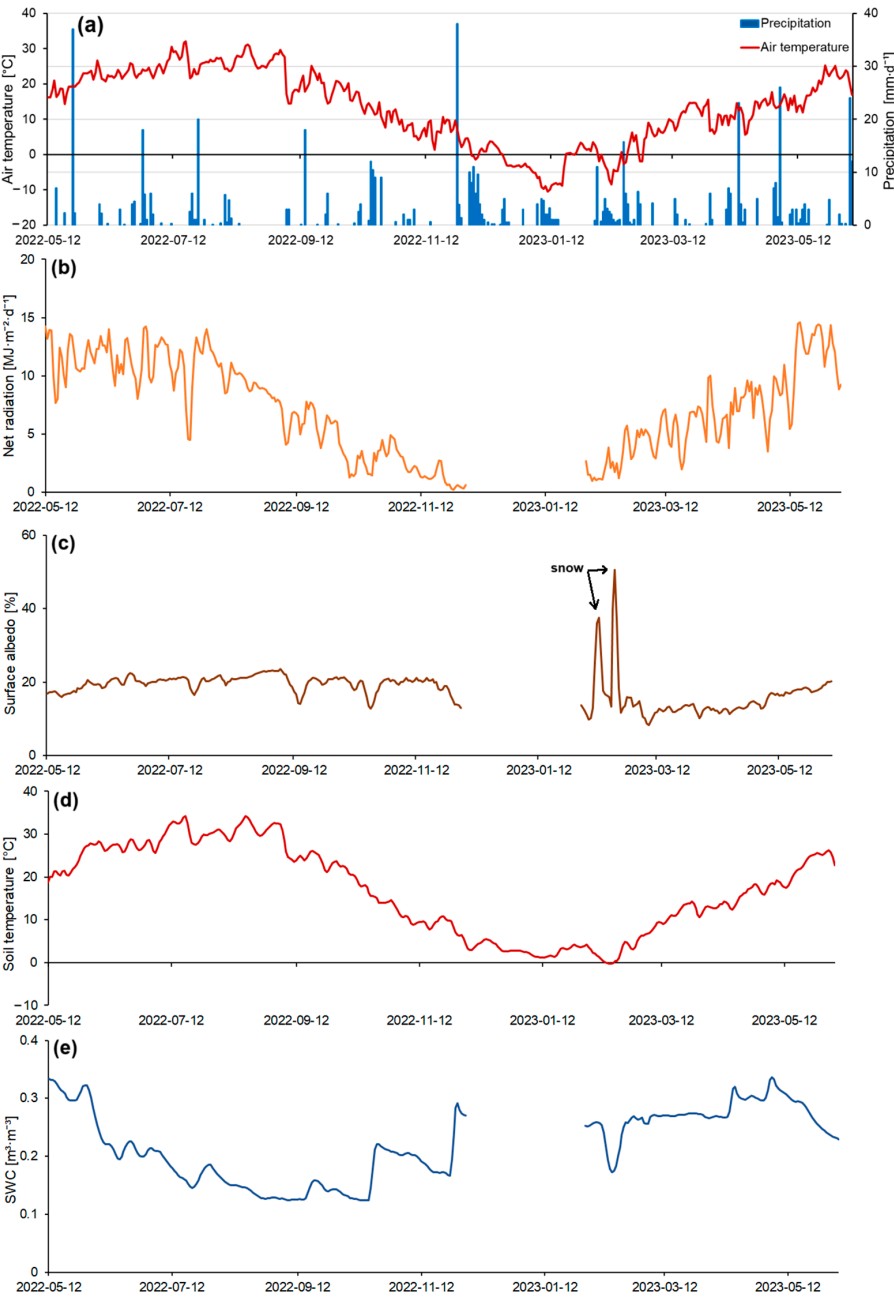

**Figure 5.** Temporal variability of mean daily temperature and daily precipitation (**a**), net radiation (**b**), surface albedo (**c**), soil temperature (**d**), and soil water content (SWC) (**e**) for the entire measurement period. Gaps in data were due to a malfunction of the measuring system during these intervals.

The seasonal variation of net radiation correlated well with the seasonal variation of the daily mean air temperature. During the study period, net radiation was determined by the incoming global solar radiation. The maximum values were observed during hot and dry weather conditions. In summer, net radiation varied from 4.5 to 14.7 MJ·m$^{-2}$·d$^{-1}$. In the first half of summer, there was a steady increase in net radiation, reaching 14.2 MJ·m$^{-2}$·d$^{-1}$ by the end of June (Figure 5b). In early August, net radiation values began to decrease to 7.7 MJ·m$^{-2}$·d$^{-1}$, which was associated with a decrease in incoming global solar radiation. In autumn, net radiation values continued to decrease from 8.15 to 0.23 MJ·m$^{-2}$·d$^{-1}$. In mid-December, the lowest daily mean net radiation values of 0.22 MJ·m$^{-2}$·d$^{-1}$ were observed. After the end of January, there was an increase in net radiation, reaching 5.4 MJ·m$^{-2}$·d$^{-1}$ by the beginning of spring.

In the spring of 2023, net radiation varied from 1.95 to 14.6 MJ·m$^{-2}$·d$^{-1}$ (Figure 5b). Local minima in net radiation values during observations occurred during periods of overcast and rainy weather. Data gaps in net radiation and albedo in December–January were due to the malfunction of instruments on the measuring mast.

Changes in net radiation were also caused by changes in the albedo of the Earth's surface. Changes in albedo were primarily related to the phenological features of the vegetation and atmospheric precipitation, leading to changes in the moisture content of the upper soil layer. In the spring and summer of 2022, after the planting of tree seedlings, the albedo ranged from 15.9 to 19.5% (Figure 5c). Albedo had clear diurnal variability and depended strongly on the wetness of the upper soil layer; the albedo of drier surface soil was higher than the albedo of wetter soil, e.g., after heavy rainfall events. Thus, during the drought in August–September 2022, the albedo reached 23.5%, mainly due to the drying out of vegetation and an increase in its reflective properties. Maximum albedo values were in winter and ranged from 35 to 50%. They were observed after snowfall. The snow cover melted almost immediately after falling. In the spring of 2023, after planting new seedlings, the albedo ranged from 11.2 to 19.3%, gradually increasing with the increase in net radiation (Figure 5b,c).

Soil temperature at the 10 cm depth at the carbon farm was fairly uniform during the entire study period (Figure 5d). Maximum temperature values were observed in July and August, reaching 34.8 °C. Further, as incoming global solar radiation decreased, soil temperature gradually decreased and reached a minimum value of about −0.3 °C in February. In spring, soil temperature gradually increased from 5.8 to 25.4 °C.

Soil water content (SWC) during the summer of 2022 gradually decreased due to insufficient precipitation and high temperatures leading to high evaporation rates. From May to August 2022, SWC decreased from 0.33 to 0.12 m$^3$·m$^{-3}$ (Figure 5e). After a dry period in September, SWC began to increase again and reached 0.29 m$^3$·m$^{-3}$ in early December. In the cold period of the year, SWC had greater variability, exceeding the variability in the warm period due to more frequent precipitation. During snowfall, SWC decreased to 0.17 m$^3$·m$^{-3}$. In the spring of 2023, there were sufficient soil moisture conditions, with SWC ranging from 0.22 to 0.33 m$^3$·m$^{-3}$. Data gaps in SWC in December–January were due to the malfunction of instruments on site.

### 3.2. Seasonal Variations of Heat Balance Components

The seasonal course of H and LE fluxes over the study period was characterized by a fairly high variability (Figure 6). The variability in heat fluxes was primarily determined by surface net radiation, soil moisture, and the structure and biophysical properties of the vegetation cover, which influence plant transpiration. Weather conditions played a significant role in regulating the transpiration of the vegetation cover in the study area.

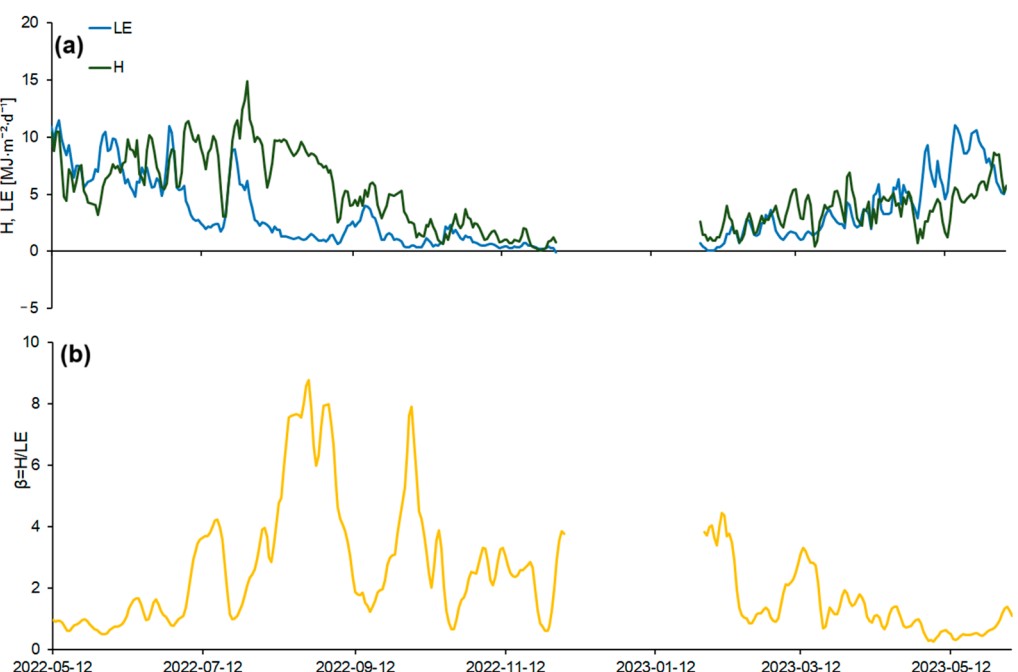

**Figure 6.** Seasonal courses of (**a**) daily sensible (H) and latent (LE) heat fluxes and (**b**) Bowen ratio (β). Gaps in data were due to a malfunction of the measuring system during these intervals.

After the melting of the snow cover and an increase in the net radiation values in the spring of 2022, there was a steady increase in H (Figure 6a). In May–early June, H values were 1.5–2 MJ·m$^{-2}$·d$^{-1}$ lower than LE due to high precipitation and sufficient moisture in the upper soil horizon. After precipitation events, the proportion of LE in the heat balance increased. Local maxima in daily LE fluxes often coincided with days of heavy precipitation. During the summer of 2022, LE flux ranged from 0.9 to 11.4 MJ·m$^{-2}$·d$^{-1}$, while H flux ranged from 3.1 to 14.9 MJ·m$^{-2}$·d$^{-1}$. The LE fluxes reached maximum values in early summer, with daily values of 11.4 MJ·m$^{-2}$·d$^{-1}$ (Figure 6a). Against the background of drying of the upper soil layer, a gradual decrease in LE and an increase in H were observed by mid-summer. The maximum H values were observed in mid-August and amounted to 14.9 MJ·m$^{-2}$·d$^{-1}$, due to vegetation drying caused by soil moisture deficits, lack of precipitation, and high average daily temperatures.

To describe the ratio of H and LE, the Bowen's ratio (β = H/LE) [23] was calculated. In late spring–early summer, β showed a stable trend towards an increase (Figure 6b). The maximum β values were observed at the end of August (β = 8.7) when H significantly exceeded LE.

In the autumn of 2022, LE flux ranged from 0.2 to 3.9 MJ·m$^{-2}$·d$^{-1}$, while H flux ranged from 0.1 to 7.5 MJ·m$^{-2}$·d$^{-1}$. B varied in this period from 0.6 to 7.8, and H was higher than LE throughout the autumn. Daily flux values gradually decreased during the autumn, reaching a minimum in December, because of declines in net radiation. Local minima of LE were observed on days when there was a large amount of precipitation (with dense clouds and low values of incoming solar radiation), as well as in winter, when the values of the net radiation were minimal. Data gaps in the fluxes during December–January were due to measurement system malfunctions. In February, H exceeded LE by an average of 0.5–1 MJ·m$^{-2}$·d$^{-1}$ (Figure 6a).

In the spring of 2023, LE ranged from 0.9 to 10.5 MJ·m$^{-2}$·d$^{-1}$ and H ranged from 0.5 to 8.6 MJ·m$^{-2}$·d$^{-1}$. From mid-April, LE began to increase and reached 11.1 MJ·m$^{-2}$·d$^{-1}$ by early June 2023 due to optimal moisture conditions and high values of incoming solar radiation, as well as transpiration and the active growth of new linden seedlings. β showed a stable decreasing trend from mid-March from 3.3 to 0.2.

The total amount of available energy that the ecosystem spent on the H and LE fluxes during the entire measurement period was 1177.4 MJ·m$^{-2}$ for LE and 1613.4 MJ·m$^{-2}$ for H. H fluxes were consistently higher than LE fluxes throughout most of the measurement period (except late spring 2023).

### 3.3. Seasonal Variations of $CO_2$ Fluxes

The seasonal dynamics of $CO_2$ flux in 2022 and 2023, according to eddy covariance measurements, were characterized by significant variability, mainly due to changes in meteorological parameters: incoming solar radiation, air temperature, precipitation, and soil moisture, as well as changes in phenological parameters and the functional activity of plant communities.

From mid-May to mid-June, daily average $CO_2$ flux at the carbon farm was close to zero (Figure 7). During periods after precipitation, the rate of $CO_2$ absorption from the atmosphere due to plant photosynthesis (gross primary productivity (GPP)) exceeded the rate of total ecosystem respiration (TER) (Figure 7). During these brief periods in late spring and early summer, the carbon farm served as a sink for $CO_2$ from the atmosphere. Maximum uptake during this period was $-0.71$ gC·m$^{-2}$·d$^{-1}$.

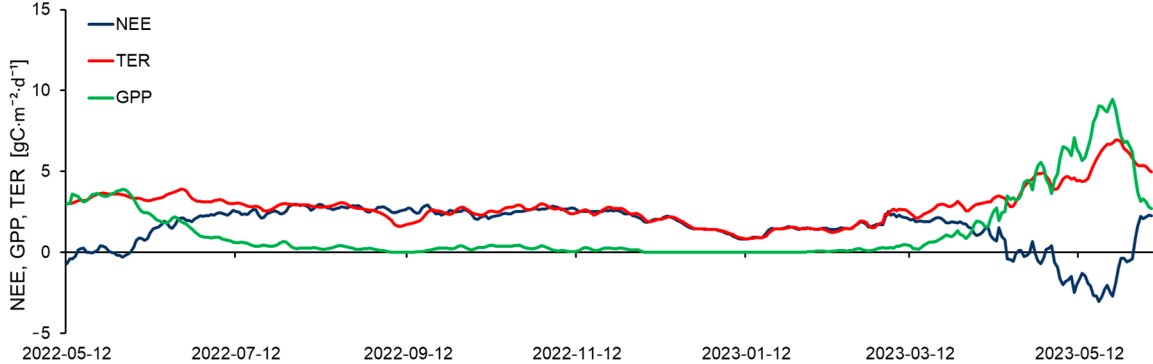

**Figure 7.** Seasonal courses of net ecosystem exchange (NEE), total ecosystem respiration (TER), and gross primary production (GPP). GPP characterizes the $CO_2$ uptake by the ecosystem.

Due to the rise in average daily temperatures to 30 °C after the end of June, and long dry periods, all previously planted vegetation on the carbon farm died. As a result, TER began to dominate in the NEE. The ecosystem primarily served as a source of $CO_2$ to the atmosphere (Figure 7). Mean daily NEE in the summer of 2022 ranged from $-0.3$ to 2.9 gC·m$^{-2}$·d$^{-1}$. Maximum emission values occurred during the longest periods of dry and hot weather (August–September), reaching 3.1 gC·m$^{-2}$·d$^{-1}$. In the autumn, as temperatures decreased, NEE decreased from 2.9 to 2.3 gC·m$^{-2}$·d$^{-1}$. By mid-January, the emission rate was 0.79 gC·m$^{-2}$·d$^{-1}$. In the cold season, temperature and soil moisture changes had the most significant impact on $CO_2$ flux, influencing soil respiration rates.

In the spring, from mid-March to the end of May 2023, NEE began to decrease from 2.5 to $-3.1$ gC·m$^{-2}$·d$^{-1}$. The ecosystem gradually became a sink of $CO_2$ from the atmosphere. The uptake of $CO_2$ during this period was facilitated by the planting of new seedlings. However, due to increasing daily temperatures and the absence of irrigation systems, the vegetation began to dry out again. The ecosystem once more served as a source of $CO_2$ to the atmosphere.

The total cumulative NEE for the entire measurement period at the carbon farm was 613.7 gC·m$^{-2}$.

During the entire observation period, local maxima of $CO_2$ flux were also observed after large amounts of precipitation. This process was observed as a result of an increase in soil moisture, which led to an acceleration of the decomposition of organic material in the soil, causing a strong increase in $CO_2$ emissions into the atmosphere (the so-called Birch effect [24]). Local minima in $CO_2$ flux were observed on average about a week after a long

period of precipitation, primarily due to increased photosynthetic activity in surviving plants [25]. The effect of temperature on the $CO_2$ flux in the study region was most strongly reflected in the dry season: during a decrease in temperature, a decrease in $CO_2$ emission was observed, mainly due to a decrease in the intensity of soil respiration. Local maxima in $CO_2$ emissions mostly coincided with hot weather periods.

Using eddy covariance measurements, NEE components were also calculated: GPP and TER (Figure 7). Throughout the observation period, mean daily GPP varied from 0 to 9.7 $gC·m^{-2}·d^{-1}$, while mean daily TER ranged from 0.8 to 7.4 $gC·m^{-2}·d^{-1}$. The calculation results showed a trend of increasing GPP and TER in late spring and early summer. Because of optimal soil moisture conditions and incoming solar radiation at the upper boundary of the vegetation cover, the integral photosynthetic productivity of the vegetation reached 3.94 $gC·m^{-2}·d^{-1}$. In early July, the highest TER values were observed, reaching 3.96 $gC·m^{-2}·d^{-1}$.

Furthermore, during the entire observation period until spring 2023, TER values were significantly higher than GPP (on average by 2.5 $gC·m^{-2}·d^{-1}$), due to the desiccation of vegetation caused by high temperatures and low precipitation. In winter, due to the almost complete absence of vegetation, GPP values tended to zero. Thus, the seasonal course of GPP was determined by the amount of photosynthesizing vegetation (green biomass) and the amount of incoming photosynthetically active radiation (PAR). In the presence of high temperatures and a lack of precipitation, soil moisture content in deep soil horizons played a significant role in influencing GPP [26,27].

In the spring of 2023, after the planting of new vegetation, GPP increased and reached maximum mean daily values for the entire observation period at 9.7 $gC·m^{-2}·d^{-1}$. However, TER also increased and mean daily values reached 7.4 $gC·m^{-2}·d^{-1}$. Subsequently, both parameters sharply decreased by June due to vegetation drying, high temperatures, and insufficient moisture.

The cumulative values of GPP were 534.9 $gC/m^2$, and TER was 1148.7 $gC/m^2$ for the entire observation period.

### 3.4. Diurnal Variations of the Energy Fluxes

To analyze the diurnal variability of H (sensible heat) and LE (latent heat) fluxes, three months were selected, representing different seasons of the year: August and October 2022 and May 2023 (Figure 8a).

In August 2022, H significantly exceeded LE at midday by an average of $160 \pm 30$ $W·m^{-2}$ (Figure 8a). The maximum daytime values for H occurred at 13 h and were $280 \pm 63$ $W·m^{-2}$, while LE peaked at 12 h with $60 \pm 31$ $W·m^{-2}$. Nocturnal H and LE fluxes were $-10.5 \pm 5.5$ $W·m^{-2}$ and $-6.4 \pm 2.5$ $W·m^{-2}$, respectively. LE was higher than H during nocturnal hours, averaging $4.2 \pm 2.5$ $W·m^{-2}$. Such dynamics were primarily due to insufficient soil moisture and high daytime air temperatures in the summer months. The mean energy balance closure in August at the carbon farm site was about 0.71.

H also continued to exceed LE during the daytime in October 2022 by an average of $47.3 \pm 19$ $W·m^{-2}$ (Figure 8a). However, the absolute maxima at midday were significantly lower than in August, at $109 \pm 63$ $W·m^{-2}$ for H and $39 \pm 26$ $W·m^{-2}$ for LE. Nocturnal H and LE fluxes were $-7.8 \pm 3.5$ $W·m^{-2}$ and $-1.5 \pm 0.5$ $W·m^{-2}$, respectively. The mean energy balance closure was lower in October than in August; in October the closure was 0.67.

In May 2023, the LE flux began to exceed the H flux, averaging $36.7 \pm 9.8$ $W·m^{-2}$ during daytime hours (Figure 8a). This was due to optimal moisture conditions (and resulting increased evaporation) and the appearance of photosynthetic green vegetation (increased transpiration of vegetation cover) at the carbon farm. Maximum LE values were observed around 13 h and reached $155 \pm 67$ $W·m^{-2}$, while H at the same time was $111 \pm 61$ $W·m^{-2}$. During the nighttime in May, LE was also significantly higher than H by $18.7 \pm 2.2$ $W·m^{-2}$. The absolute nighttime minima in May 2023 were $0.17 \pm 0.07$ $W·m^{-2}$ for LE and $-21 \pm 7.2$ $W·m^{-2}$ for H. Minimum LE values were observed in the early morning

hours at 3–4 h, while H was at its minimum in the first half of the night from 20–23 h. The mean energy balance closure in May was 0.82.

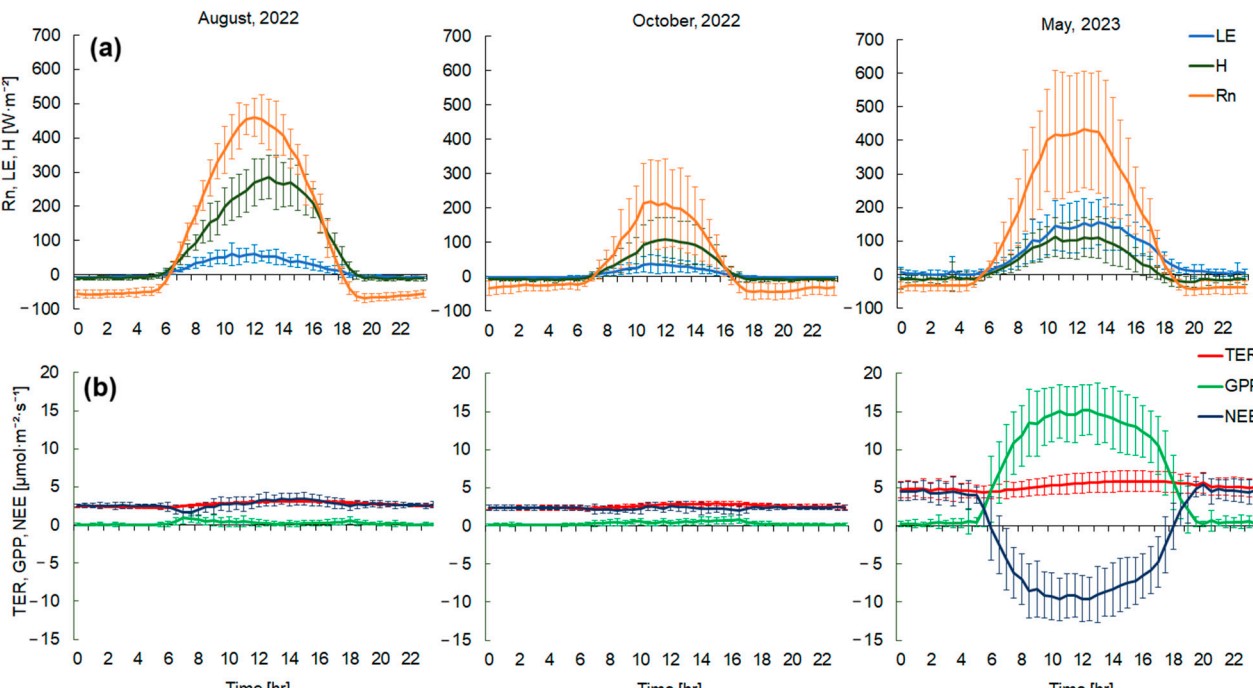

**Figure 8.** Diurnal courses of (**a**) the mean energy balance components and (**b**) the mean $CO_2$ balance components for three months (August, October 2022, and May 2023). Vertical whiskers indicate standard deviations (SD). Negative NEE denotes carbon uptake by the ecosystem. GPP characterizes the $CO_2$ uptake by the ecosystem.

### 3.5. Diurnal Variations of $CO_2$ Fluxes

The diurnal variation of $CO_2$ showed significant variability throughout the entire observation period (Figure 8b). This variability was primarily influenced by incoming solar radiation, soil and air temperatures, vegetation structure, and moisture conditions.

Due to the almost complete drying of vegetation in August 2022, $CO_2$ emissions into the atmosphere were observed during both the day and at night. The highest NEE values ($3.47 \pm 0.7 \ \mu mol \cdot m^{-2} \cdot s^{-1}$) occurred from 13–15 h (Figure 8b) when air and soil temperatures were at their maximum. By evening, with decreasing temperatures, NEE also decreased to $2.4 \pm 0.6 \ \mu mol \cdot m^{-2} \cdot s^{-1}$. The minimum NEE values were observed in the morning from 7–9 h, reaching $1.6 \pm 0.5 \ \mu mol \cdot m^{-2} \cdot s^{-1}$ (Figure 8b). During the day in August, NEE was primarily driven by soil respiration and the decomposition of woody and herbaceous vegetation residues due to insufficient moisture conditions and high daily temperatures. The diurnal course of GPP had two distinct peaks, in the morning and evening hours ($0.9 \pm 0.5 \ \mu mol \cdot m^{-2} \cdot s^{-1}$ and $0.5 \pm 0.2 \ \mu mol \cdot m^{-2} \cdot s^{-1}$). In the morning hours, the increase in GPP was determined by an increase in incoming solar radiation. In contrast, at midday, there was a significant depression in GPP ($0.1 \pm 0.04 \ \mu mol \cdot m^{-2} \cdot s^{-1}$) (Figure 8b) associated with moisture deficits in the root zones of plants and thermal stress, leading to stomatal closure. TER was mainly influenced by temperature changes. The TER maximum was observed in the late afternoon in August 2023, reaching $3.4 \pm 0.7 \ \mu mol \cdot m^{-2} \cdot s^{-1}$.

In October 2022, $CO_2$ emissions were still observed during both daytime and nighttime. The diurnal course of $CO_2$ flux components was similar to August. The maximum NEE values were observed at 12–13 h (Figure 8b) and amounted to $2.6 \pm 0.5 \ \mu mol \cdot m^{-2} \cdot s^{-1}$. The minimum NEE was at 8 h and was $2.0 \pm 0.5 \ \mu mol \cdot m^{-2} \cdot s^{-1}$. In October, the $CO_2$ flux was also primarily determined by soil respiration and the decomposition of vegetation residues. The maximum TER was observed at 14–16 h and amounted to $2.9 \pm 0.3 \ \mu mol \cdot m^{-2} \cdot s^{-1}$. GPP

gradually increased throughout the day, reaching maximum values of $0.8 \pm 0.5$ $\mu$mol·m$^{-2}$·s$^{-1}$ at 16 h.

In May 2023, due to active plant growth, the carbon farm served as a sink of $CO_2$ from the atmosphere during the daytime. NEE reached minimum values of $-9.5 \pm 2.8$ $\mu$mol·m$^{-2}$·s$^{-1}$ at 11–12 h (Figure 8b). At the same time, there were maximum GPP values of $15.2 \pm 3.2$ $\mu$mol·m$^{-2}$·s$^{-1}$. Under optimal moisture and temperature conditions, and with increasing incoming solar radiation at the upper boundary of the plant cover, the integral photosynthetic productivity of the vegetation increased. TER was also influenced by temperature changes, with maximum values observed in the late afternoon at 14–16 h, amounting to $5.8 \pm 1.3$ $\mu$mol·m$^{-2}$·s$^{-1}$. During the nighttime, the carbon farm served as a source of $CO_2$ as it depended only on ecosystem respiration. Maximum NEE values were at 20–22 h and amounted to $4.8 \pm 1.0$ $\mu$mol·m$^{-2}$·s$^{-1}$.

## 4. Discussion

As a result of this study, new experimental data on GHG fluxes from 2022 to 2023 were obtained and analyzed at the anthropogenically disturbed site in the Chechen Republic. It should be noted that this is the first case in Russia to conduct this kind of observation in such a reforestation area. In the global scientific community, similar experiments have also been sporadic. There is an urgent need to continue such experimental studies in different parts of the world to analyze the interaction between the atmosphere and the underlying surface, taking into account the existing heterogeneity and complexity of its landscape structure. The experimental data obtained in the course of such studies are important for understanding the mechanisms of interaction between terrestrial ecosystems and the atmosphere. Long-term data series from different ecosystems also will allow us to study in detail the dependence of greenhouse gas fluxes on environmental factors, as well as provide more detailed estimates of carbon stocks in different ecosystems. The model approaches developed based on long-term data analysis of various levels of complexity and scale should, in the future, enable an adequate description and ultimately the prediction of the variability of carbon exchange and heat exchange in different types of ecosystems with varying degrees of anthropogenic impact. In combination with satellite measurements, such monitoring is able to improve regional carbon balance estimates and carbon stocks.

### 4.1. Heat Fluxes

A comparative analysis of H and LE fluxes at the carbon farm over the entire measurement period showed that the prolonged hot and dry weather established from the end of July 2022 led to severe drying out of the soil, the death of previously planted vegetation, and consequently, a gradual decrease in LE and an increase in H. Until April 2023, the H flux exceeded the LE flux. Throughout the second half of 2022, the anthropogenically disturbed ecosystem was unable to recover. Vegetation recovery rates following disturbances and moisture conditions are important factors controlling evaporation. Local weather conditions and low dew rates can increase daily H fluxes and reduce LE fluxes. Similar results regarding the dynamics of H and LE fluxes in forest restoration areas can be observed in studies focused on clear-cut areas in forests [28–30].

After the death of almost all vegetation in August–September 2022, evapotranspiration was determined primarily by soil surface evaporation and depended on the moisture content of the upper soil horizon. Similar results were obtained by Zha et al. [31] in a two-year clear-cut in jack pine in Saskatchewan. After the replanting of *Tilia* saplings in May–April 2023, LE increased again and was regulated by the vegetation cover. During this period, due to sufficient precipitation providing optimal soil moisture conditions, ecosystem recovery occurred relatively quickly after anthropogenic disturbances. Similar intensive recovery after anthropogenic disturbances (after clear-cutting of forests) was described in the works of Humphreys et al. [32] and Paul-Limoges et al. [33].

As a result, weather conditions played a decisive role in regulating the transpiration of the vegetation cover in the study area. Local minima in the net radiation, H, and LE

fluxes during the warm season were observed on rainy days. Maximum energy flux values occurred during warm and sunny weather. Thus, H and LE fluxes were directly dependent on the surface net radiation, which in turn was regulated by the surface albedo. Albedo was characterized by a clear daily variability and depended on the moisture content of the upper soil layer. Similar relationships characterizing the decrease in albedo during rainy periods were obtained by McCaughey [34] and Mamkin et al. [6] in clear-cut areas. Local albedo maxima of the Earth's surface were observed during the presence of a stable snow cover.

*4.2. Carbon Dioxide Fluxes*

Analysis of $CO_2$ fluxes at the carbon farm based on eddy covariance measurements showed that, for almost the entire observation period (except May to early June 2022 and April to May 2023), positive $CO_2$ fluxes were observed, meaning that $CO_2$ emissions into the atmosphere exceeded $CO_2$ uptake by the Earth's surface. The significant loss of photosynthesizing biomass due to drought led to a considerable reduction in GPP and, as a result, an increase in NEE. Similar results can be seen in studies by Amiro et al. [35], Mamkin et al. [6] and some others [33,36], where the main changes in the $CO_2$ balance in anthropogenically disturbed ecosystems were caused by decreases in GPP. The average emission values over the entire observation period were around 2.5 $gC \cdot m^{-2} \cdot d^{-1}$. The planting of new seedings in the spring of 2023 made a significant contribution to the $CO_2$ balance during the study period. In April–May 2023, there were favorable weather conditions for active development and restoration of vegetation. During this period, the highest GPP values of 9.7 $gC \cdot m^{-2} \cdot d^{-1}$ were observed, due to high air temperatures and large amounts of precipitation. Precipitation, in turn, provided optimal conditions for soil moisture.

Despite the presence of periods of $CO_2$ uptake, the total cumulative NEE for the entire period (613 $gC \cdot m^{-2}$) still indicated that the carbon farm was a source of $CO_2$ into the atmosphere. For comparison with other sites, the following cumulative values of NEE were observed in the first year of forest restoration: 633.6 $gC \cdot m^{-2}$ in a clear-cut area in the southern taiga in Russia [6], from 1270 to 200 $gC \cdot m^{-2}$ in boreal ecosystems in Florida [35], 1000 $gC \cdot m^{-2}$ in a clear-cut area on Vancouver Island [33]. In some studies, conducted on sites that were (like the carbon farm) or are currently landfills, it is mentioned that decomposing waste residues under certain weather conditions can also be sources of $CO_2$ and other GHGs into the atmosphere [37]. In the study by Hegde, U. et al. [38], it is shown that landfills aged 2–3 years have the highest $CO_2$ emissions compared with a 5-year-old landfill. Although the carbon farm is a reclaimed landfill that has been properly cleaned up, decomposing landfill residues can also contribute to $CO_2$ emissions into the atmosphere.

However, reforestation is one of the most promising methods to increase the ability of natural ecosystems to absorb GHGs and reduce their emissions into the atmosphere. Several studies that also involved planting vegetation on anthropogenically disturbed areas have shown that the carbon uptake capacity of newly planted vegetation increases by 2–3 years after planting. In the study by Zona et al. [39], immediately after tree planting, there was stable $CO_2$ emission into the atmosphere from June to December ($2.76 \pm 0.16$ Mg $CO_{2eq}$ $ha^{-1}$). However, in the second year, the plantation already showed net $CO_2$ uptake ($-3.51 \pm 0.56$ Mg $CO_{2eq}$ $ha^{-1}$). It was also noted that water availability was an important factor in controlling GHG emissions at the plantation. In another study by Cabral et al. [40], the plantation absorbed $CO_2$ from the atmosphere in both the first and second years ($-7643.0 \pm 129.0$ and $-4615.0 \pm 124.0$ $gCO_{2eq} \cdot m^{-2} \cdot yr^{-1}$, respectively), but all this time it was a source of other GHGs into the atmosphere. The carbon farm became a source of $CO_2$ into the atmosphere due to drought in the first year after tree planting. However, under optimal temperatures and soil moisture conditions, vegetation on the site is expected to absorb carbon dioxide from the atmosphere over the next few years.

## 5. Conclusions

As a result of this study, new experimental data on greenhouse gas fluxes at the reforestation site of the Chechen carbon polygon were obtained and analyzed. The results of meteorological and eddy covariance flux measurements showed significant variability in heat and $CO_2$ fluxes in the first year after vegetation planting at the carbon farm. Certain patterns in the radiation, energy, and carbon balances were also identified. The carbon farm served as a source of $CO_2$ into the atmosphere for a significant part of the time (except for May–June 2022 and April–May 2023) due to the death of almost all planted seedlings as a result of prolonged hot and dry weather conditions in the summer of 2022. The absorption of $CO_2$ by the ecosystem was observed only due to photosynthetic vegetation during periods of sufficient soil moisture. Sensible and latent heat fluxes at the carbon farm also depended mainly on weather conditions.

This experimental study conducted to determine the temporal variability of greenhouse gas fluxes on the anthropogenically disturbed site of the carbon farm in the Chechen Republic is extremely important for solving various meteorological and environmental problems, including the development and improvement of climate forecasting accuracy and carbon footprint reduction. Given the diversity of factors influencing heat and $CO_2$ transport, the results of this work should be considered as a specific case within the complex of studies on energy and mass exchange processes in reforestation ecosystems.

**Author Contributions:** Conceptualization, E.S. and J.K.; methodology, E.S. and J.K.; software, E.S. and N.M.; validation, E.S., N.M., L.M. and J.K.; formal analysis, E.S. and N.M.; investigation, E.S.; resources, N.M.; data curation, N.M.; writing—original draft preparation, E.S.; writing—review and editing, J.K.; visualization, E.S. and N.M.; supervision, J.K.; project administration, L.M.; funding acquisition, L.M. All authors have read and agreed to the published version of the manuscript.

**Funding:** This research was funded by state assignment of the Ministry of Science and High Education of the Russian Federation FZNU-2023-0001.

**Data Availability Statement:** The data obtained in the study can be requested from the authors at justemail552@mail.ru.

**Acknowledgments:** The authors express their sincere acknowledgment to Vitaly Avilov for the installation and support of the measuring complex in the Chechen Republic.

**Conflicts of Interest:** The authors declare no conflict of interest.

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
