# Peer review of "Carbon Dioxide and Heat Fluxes during Reforestation in the North Caucasus"

_forests, doi:10.3390/f14122368_

Round 1
Reviewer 1 Report
Comments and Suggestions for Authors
The manuscript "Carbon dioxide and heat fluxes during reforestation in the North Caucasus" contributes to environmental science and forestry. The research presented in this paper offers significant insights into the seasonal variability of greenhouse gas fluxes in reforestation ecosystems. The subject matter is indeed of great interest, and it has the potential to enhance our understanding of the complex interactions between terrestrial ecosystems and the atmosphere. For that, please bring some numbers typically observed in other environments for comparisons.
One important question that may arise from this research is the need to increase the timeline of measurements for long-term monitoring. Otherwise, the results may not be conclusive. This could be better stated in the research (limitation).
While the study provides valuable data on the sensitivity of greenhouse gas fluxes to environmental factors, it would be interesting to explore the benefits of extending the measurement duration. This question could be addressed in the discussion section by considering the potential advantages of conducting measurements over longer periods in terms of comprehensiveness and cost-effectiveness. Specifically, how might longer-term monitoring provide a more comprehensive understanding of the ecosystem's response to changing environmental conditions and human activities, and what cost implications might be associated with such an extension?
Additionally, the paper mentions the use of various devices and approaches. It would be valuable to include a dedicated subtopic in the discussion section that addresses the costs associated with the acquisition of such devices, including handling them (and maintenance) and how they may impact future research, mainly in remote areas where power supply and eventual damage risk are involved, requiring local people to support such research initiatives. This would help readers understand the practical aspects of implementing similar measurements in different ecosystems and the financial considerations involved.
These questions could enrich the manuscript and provide valuable insights into the long-term monitoring potential and practical considerations for similar research endeavors. Considering long-time monitoring, please detail how to perform it and what could be linked with local forest inventories and satellite measurements to enhance the paper even more.
Reviewer 2 Report
Comments and Suggestions for Authors
Manuscript 'Carbon dioxide and heat fluxes during reforestation in the North Caucasus'
line 107 'plantings of various deciduous tree species, including mulberry, linden, willow, poplar, and ash, were conducted.' - please state the proportion of each species and planting density
line 113 'To measure greenhouse gas fluxes at the Carbon Farm site in the Chechen Republic, standardized for FLUXNET sites measuring equipment is used, including a meteorological station and an eddy covariance - station for measuring CO2 flux using the eddy covariance method (Figure 2) [13]. The height of the flux tower is 3.1 m, it is located in the central part of the Carbon Farm in an area planted with linden seedlings.' - my understanding of this is and from Figure 3 is that there is one sample site in the centre of the planted area? this raises concern about adequate replication
line 205 'During the study period from May 12, 2022, to June 7, 2023' - there is a single years data? this also raises concern about adequate replication
there appear to be no statistical analyses of e.g. correlations between CO2 flux and climatic variables, difficult because of the limited replication both temporally and spatially
line 114 'standardized for FLUXNET sites measuring equipment' - are there other sites using this equipment? if so the data they have collected needs to be included in order to provide some form of replication. Further, multiple years need to be included in order to provide reliable data.
Reviewer 3 Report
Comments and Suggestions for Authors
Dear Authors,
I have reviewed the paper titled: “Carbon dioxide and heat fluxes during reforestation in the North Caucasus". In my opinion, the aims of the paper are germane with “Forests” journal topic, however in the present form, the paper has some important flaws. The contribution of this paper to the scientific knowledge is good. I understand the difficult work done, but as a reviewer it is my duty to highlight the gaps in order to improve the research approach and its presentation to the international scientific community. Please, in order to have a good possibility for a possible future publication I attached the pdf file with some important comments. Further I suggest you to improve the statistics showed in the paper and to pay particular attention to make a better linkage among aims and conclusions.

Comments on the Quality of English LanguageOnly some minor mistakes are present in the paper referred to the english grammar, I suggest a further light editing.
Reviewer 4 Report
Comments and Suggestions for Authors
This study used new experimental data on greenhouse gas fluxes seasonal variability to assess their sensitivity to changes in environmental conditions. Forest restoration and land reclamation are essential components of the low-carbon development and decarbonization strategy of the World economy. However, the authors should modify the following issues:
(1) The research significance of this study should be supplemented in the abstract;
(2) The authors should clearly list the specific research objectives of this study;
(3) The authors should explain the meaning of the abbreviations in Figure 4;
(4) What are the sources of long-term monthly meteorological data?
(5) The corresponding figure should preferably appear in the position where this figure name first appears in the main text, such as Figure 5;
(6) What does the column in Figure 5a represent? It's difficult to understand the meaning of this figure;
(7) There are some unclear expressions in the results. For example, Lines 401The mean energy balance closure was somewhat lower in October than in August: 0.67. What does the author mean? The authors should reorganize the results section;
(8) Why did the authors choose these three months to represent different seasons for analysis? Using monthly averages from different seasons seems more accurate;
(9) The expression of the conclusion is somewhat confusing. The authors should refine the main conclusions.
Comments on the Quality of English LanguageThe quality of English language can be improved.
Round 2
Reviewer 1 Report
Comments and Suggestions for Authors
The comments were properly added in the manuscript and I am favorable to recommended it for publication.
Author Response
We thank the reviewer for their helpful and constructive comments and recommendations.
Reviewer 2 Report
Comments and Suggestions for Authors
Thank you for the clarification. The manuscript describes a baseline data collection year, in its current state it would be better submitted as a short communication rather than a research article given the current limitations on the data collected.
Author Response
The authors would like to express their sincere gratitude to the reviewer for the careful reading of our article and the suggestions made. The authors are aware that the scientific community currently has access to a large number of scientific databases and publications where extensive material on the climate-regulating functions of terrestrial ecosystems, including forests, can be found.
However, the authors hope that the material presented in the format of this article will allow the interested reader to become acquainted with the region and area of study, the experimental setup, and more detailed results of the work. These are the first measurements of ecosystem greenhouse gas and heat fluxes using the eddy covariance method in the North Caucasus presented in English-language scientific literature. The Caucasus plays a significant role in the climate of southern Russia, and the region possesses unique natural conditions. The Chechen Republic has only begun to develop a program for experimental ecological research; this is the first major project in the region using modern scientific equipment.
The authors keep track of works in the field of their scientific interests. We are pleased to learn from the results of experimental observations, realizing that researchers do not always manage to obtain long-term data series in different regions of the world. For instance, the research results mentioned below are based on short-term observations (no more than one year).
Zona, D., Janssens, I. A., Aubinet, M., Gioli, B., Vicca, S., Fichot, R., & Ceulemans, R. (2013). Fluxes of the greenhouse gases (CO2, CH4 and N2O) above a short-rotation poplar plantation after conversion from agricultural land. Agricultural and Forest Meteorology, 169, 100-110. http://dx.doi.org/10.1016/j.agrformet.2012.10.008
Mamkin, V., Kurbatova, J., Avilov, V., Ivanov, D., Kuricheva, O., Varlagin, A., ... & Olchev, A. (2019). Energy and CO2 exchange in an undisturbed spruce forest and clear-cut in the Southern Taiga. Agricultural and Forest Meteorology, 265, 252-268. https://doi.org/10.1016/j.agrformet.2018.11.018
Deng, Z.; Liu, X.; Zu, H.; Luo, J.; Chen, Y.; Yi, M.; Wang, X.; Liang, X.; Zhang, X.; Yan, W. Spatial and Temporal Variations of Carbon Dioxide Fluxes in Urban Ecosystems of Changsha, China. Forests 2023, 14, 2201. https://doi.org/10.3390/f14112201
Zheng, X.; Yang, F.; Mamtimin, A.; Huo, X.; Gao, J.; Ji, C.; Abudukade, S.; Li, C.; Sun, Y.; Wang, W.; et al. Farmland Carbon and Water Exchange and Its Response to Environmental Factors in Arid Northwest China. Land 2023, 12, 1988. https://doi.org/10.3390/land12111988
Badraghi, A.; Novotná, B.; Frouz, J.; Krištof, K.; Trakovický, M.; Juriga, M.; Chvila, B.; Montagnani, L. Temporal Dynamics of CO2 Fluxes over a Non-Irrigated Vineyard. Land 2023, 12, 1925. https://doi.org/10.3390/land12101925
We hope to continue our work over an extended period, which will enable us to present to the readers both the interannual variability of heat and greenhouse gas fluxes.
Reviewer 3 Report
Comments and Suggestions for Authors
Dear Authors,
I have reviewed for the second round the paper titled: “Carbon dioxide and heat fluxes during reforestation in the North Caucasus". In my opinion, the aims of the paper are germane with “Forests” journal topic, in the present form, the paper has only some minor flaws. The contribution of this paper to the scientific knowledge is good. I understand the difficult work done, but as a reviewer it is my duty to highlight the gaps in order to improve the research approach and its presentation to the international scientific community. Please, in order to have a good possibility for a possible future publication I suggest to check the abstract, that still lacks take home message. These are some major findings that can be helpful for forest managers attainable from your study... I suggest to insert them in the final part of the abstract.
Author Response
Thank you for your comment. In the final part of the abstract we have included the main results that may be useful for forest managers and other researchers. (page 1, line 19-29) “One year of eddy covariance flux measurements (May 2022 – June 2023) demonstrated that CO2 uptake by the reforestation area in a humid continental climate with hot summers and cold winters is determined by the moisture conditions of the growing season. The cumulative Net Ecosystem Exchange (NEE) for the entire measurement period at the Carbon Farm was 613.7 gC·m-2. The uptake of CO2 by the reforestation area was observed only due to active seedlings growth during periods of sufficient soil moisture (May 2023). During this time, total NEE uptake was 48.7 gC·m-². Sensible and latent heat fluxes were also dependent on weather conditions, primarily on incoming solar radiation and moisture conditions. For the successful implementation of climate projects aimed at the creation of artificial forest ecosystems with high CO2 uptake capacity, it is essential to develop appropriate hydro-meliorative measures that ensure a sufficient amount of available soil moisture.”
Reviewer 4 Report
Comments and Suggestions for Authors
The authors have changed most of our concerns. However, this is an issue that requires further explanation and discussion, namely Figure 7, why is NEE greater than GPP?
Author Response
Thanks for your question. NEE (Net Ecosystem Exchange) is indeed greater than the GPP (Gross Primary Production) most of the time. GPP and NEE are related by the equation: NEE= TER -GPP. GPP is the rate at which solar energy is captured in sugar molecules during photosynthesis. During August-September 2022, a drought occurred, leading to the death of a large proportion of the planted photosynthetic vegetation. Consequently, the GPP share in the total ecosystem exchange significantly decreased, and NEE was primarily comprised of the ecosystem's Total Ecosystem Respiration (TER). Due to this event, up until the spring of 2023, prior to the planting of new tree saplings, NEE was higher than GPP.